# Development of an RNA aptamer-assisted CRISPR/Cas9 system for efficiently generating and isolating Cas9-free mutants in plant

Sha Liu[1]⊜, Jiuyuan Bai[1,2]⊜*, Bo Zhan[1], Junyu Yao[1], Jiayu Zhang[1], Jia Yi[1], Mengyue Dong[1], Qicong Li[1], Yucheng Shen[1], Yazhou Chen[3], Yun Zhao[1]*

**1** Key Laboratory of Bio-Resource and Eco-Environment of Ministry of Education, College of Life Sciences, Sichuan University, Chengdu, China, **2** Microbiology and Metabolic Engineering Key Laboratory of Sichuan Province, College of Life Sciences, Sichuan University, Chengdu, China, **3** Hubei Hongshan Laboratory, Wuhan, China

⊜ These authors contributed equally to this work.
* zhaoyun@scu.edu.cn (YZ); Baijiuy@scu.edu.cn (JB)

## Abstract

The CRISPR/Cas9 gene-editing system is a powerful tool in plant genetic engineering; however, screening for Cas9-free edited plants remains complex and time-consuming. To address this limitation, we developed an RNA aptamer-assisted CRISPR/Cas9 system, termed 3WJ-4×Bro/Cas9. In this system, the engineered RNA aptamer 3WJ-4×Bro functions as a transcriptional reporter, serving as an alternative to traditional fluorescent proteins and thus avoiding their potential interference with Cas9 activity. Compared to the conventional GFP/Cas9 system, 3WJ-4×Bro/Cas9 showed more efficient transformation and higher accuracy in fluorescence-based selection of positive $T_1$ transformants, without significantly affecting plant growth. Furthermore, 3WJ-4×Bro/Cas9 achieved a 78.6% increase in the $T_1$ mutation rate compared to GFP/Cas9, with the homozygous mutation rate reaching 1.78%. In addition, 3WJ-4×Bro/Cas9 enabled fluorescence-based visual screening in the $T_2$ generation for rapid identification of Cas9-free mutants, improving sorting efficiency by 30.2% over the GFP-based method. Moreover, 3WJ-4×Bro/Cas9 enabled more efficient generation of homozygous double-target mutants compared to GFP/Cas9. These results demonstrate that the 3WJ-4×Bro/Cas9 system provides a non-transgenic, efficient, and broadly applicable strategy for plant genome editing and selection.

## Author summary

While gene editing has become an essential tool for plant research and crop improvement, traditional fluorescent protein-assisted CRISPR/Cas9 low accuracy in screening Cas9-free mutants. In this study, we aimed to address the challenge of screening for Cas9-free edited plants in the CRISPR/Cas9 gene editing

**Data availability statement:** All data supporting the conclusions of this study are available in the main text and in the Supplementary Materials (S2_Table). All materials (seed stocks, plasmids) are available from the lab (mlwang@scu.edu.cn) upon reasonable request.

**Funding:** This research was funded by the National Natural Science Foundation of China, grant numbers 32201241 and 32171457. The funders played no role in the study design, data collection, analysis, decision to publish or preparation of the manuscript.

**Competing interests:** The authors have declared that no competing interests exist.

system. To overcome this limitation, we developed an RNA aptamer-assisted CRISPR/Cas9 system, termed 3WJ-4×Bro/Cas9, which utilizes an engineered RNA aptamer as a transcriptional reporter for fluorescence-based selection of positive transformants and Cas9-free mutants. Our results show that the 3WJ-4×Bro/Cas9 system outperforms the conventional GFP/Cas9 system in terms of transformation efficiency, mutation rate, and Cas9-free line identification in *Arabidopsis thaliana*. This method holds promise to accelerate advances in plant genetics research and support future crop improvement efforts.

## Introduction

In recent years, gene editing technologies-particularly the CRISPR/Cas system-have revolutionized crop genetic improvement and functional genomics research, owing to their high efficiency, precision, and operational simplicity. In major crops such as rice [1–4], wheat [5], potato [6], and tomato [7], CRISPR/Cas has been successfully applied to enhance disease resistance, increase yield, and regulate fruit development. These advancements have significantly accelerated the breeding of elite cultivars and highlighted the transformative potential of gene editing in plant biotechnology.

However, despite these promising applications, key challenges remain in the practical implementation of gene editing technologies, particularly in achieving genetic stability and transgene-free edited lines. Continuous expression of editing components such as Cas9 may result in off-target effects, genome instability, and the retention of transgenic markers [8,9], all of which hinder the broader use of edited lines in commercial breeding and downstream functional studies [10]. Consequently, there is an urgent need to develop strategies that can efficiently generate Cas-free plants-those devoid of exogenous editing elements-without compromising plant growth or developmental integrity. A simple and effective Cas-free selection system would greatly facilitate the transition of CRISPR-based technologies from the laboratory to field applications.

Several strategies have been proposed for Cas-free plant selection, including fluorescent reporter systems [11,12], gamete or zygote lethality-based selection [13], and genetic segregation of edited alleles [14]. Fluorescent markers, such as GFP or RFP, allow intuitive identification of transgenic plants by tagging Cas9 expression [15–18]; however, these approaches often require co-expression of additional proteins, which may interfere with editing efficiency and introduce long-term expression burdens. Alternatively, lethality-based strategies eliminate transgenic individuals through embryo lethality [14,19] or reproductive disruption [20]. Although this method offers high screening efficiency, it frequently results in embryo lethality, abnormal growth, and reduced transformation efficiency, limiting its applicability across diverse plant species. Other approaches, such as resistance marker screening [21], high-throughput PCR, or Cas9 protein detection, can be technically demanding, resource-intensive, and unsuitable for large-scale applications [22–24].

 PLOS Genetics

To overcome these limitations, RNA-based systems are emerging as a compelling alternative for gene editing regulation and selection. Compared to irreversible DNA-level modifications, RNA-level regulation offers greater flexibility, dynamic responsiveness, and programmability. These features make RNA systems a compelling alternative for efficient and precise gene-editing selection. Among RNA-based tools, RNA aptamers are short, structured RNA sequences capable of binding specifically to small-molecule fluorescent ligands, producing fluorescence upon interaction [25,26]. Aptamers such as Bro [27], Spinach [25], and Mango [28] have been widely used in mammalian and yeast systems for applications such as mRNA tracking, transcriptional regulation, and synthetic biology, owing to their excellent spatiotemporal resolution and low background noise [29–31]. Although their application in plant systems is still nascent, initial studies have shown promise for live-cell imaging and in vivo mRNA detection [32]. Compared with traditional fluorescent protein reporters, RNA aptamers are smaller, possess well-defined secondary structures, function independently of translation, and are less likely to interfere with endogenous protein function, thereby offering greater compatibility with plant gene-editing systems and enabling real-time monitoring capabilities.

Here, we introduce and optimize a novel RNA aptamer-based CRISPR/Cas9 reporter system for plant applications. By fusing RNA aptamers to Cas9 transcripts, we enable direct visualization and regulation of Cas9 expression at the RNA level, without impairing its gene-editing function. This system not only improves fluorescence intensity and editing efficiency compared to conventional GFP-based systems, but also facilitates rapid identification of Cas-free edited lines. Our work expands the application of RNA aptamers in plants and provides a promising strategy for precise regulation and visual screening in genome editing, with substantial theoretical and practical value.

## Results

### *In vitro* fluorescence characterization of RNA aptamers

To improve fluorescence intensity of 3WJ-4×Bro RNA aptamer we reported previously, the polymerized RNA aptamers were generated by fusing double or triple 3WJ-4×Bro in a 48-nt F30 scaffold, termed respectively 3WJ-8×Bro and 3WJ-12×Bro (S1 Fig). These aptamers are none fluorescence unless binding the cognate dye DFHBI-1T in plant cells, enabling specific RNA imaging (Fig 1a). *In vitro* fluorescence imaging of the polymerized RNA aptamers showed an increasing signal intensity with the number of Bro units (Fig 1b), consistent with quantitative results (Fig 1c), indicating the multimerization is an effective strategy to create fluorescence-enhanced RNA aptamers.

Next, photostability analysis under continuous ambient light exposure revealed that 3WJ-12×Bro displayed the fastest fluorescence decay, followed by 3WJ-4×Bro, while 3WJ-8×Bro demonstrated the slowest decay (Fig 1d). The minimum $T_m$ of the aptamer is 58 degrees, indicating that its folded conformation is thermodynamically stable at physiological temperature (Fig 1e). Besides, we found that the polymerized RNA aptamers achieved fluorescence saturation at 60 mM $K^+$ and 8 mM $Mg^{2+}$, indicating a relative low ion-dependence (Fig 1f and 1g). We subsequently performed a RNase A digestion assay to assess enzymatic stability, a crucial factor for RNA aptamer to preserve fluorescence integrity *in vivo* contexts. All polymerized RNA aptamers showed equivalent degradation kinetics, with comparable resistance to enzymatic cleavage (Fig 1h). These characteristics demonstrated the polymerized RNA aptamers possess potential in transcriptional reporting of gene expression *in vivo* cells.

### Rationale design of RAA/Cas9

To facilitate rapid identification of positive transformants and Cas9-free mutants, Current CRISPR/Cas9 system usually employ fluorescent proteins as reporters in frame fusion with *Cas9* via a self-cleavage 2A peptide [19–20]. While these approaches enable fluorescence-based selection of $T_1$ transformants and $T_2$ mutants, their capacity to generate transgene-free mutants remains suboptimal, which is possibly attributed to the activity-limited Cas9 caused by incomplete cleavage of the 2A peptide. [19]. To overcome constraint of protein-based reporter, we developed an RNA aptamer-assisted CRISPR/

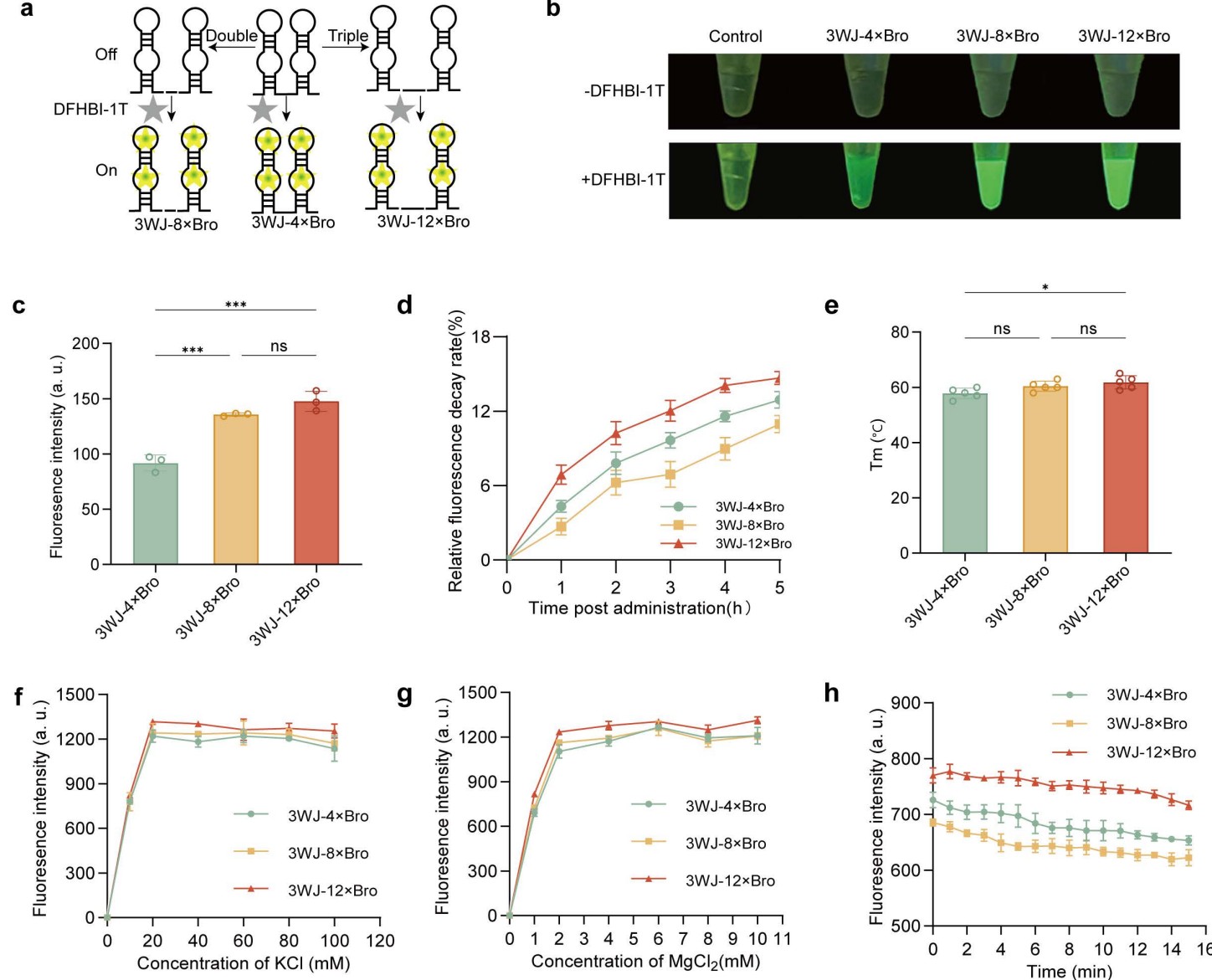

**Fig 1. *In vitro* fluorescence imaging and physicochemical characterization of the 3WJ-n×Bro fluorescent aptamer. (a)** Schematic view of polymerization of RNA aptamers. Each construct consists of a 3WJ scaffold (black lines) carrying multiple Bro units (protrusions) that bind DFHBI-1T to generate fluorescence; 3WJ-8×Bro and 3WJ-12×Bro were generated by tandemly repeating Bro units on the 3WJ-4×Bro scaffold. **(b)** In vitro fluorescence imaging of the RNA aptamers. **(c)** quantification of fluorescence intensity in **(b)**. Error bars indicate mean±SD (n=3). Statistical significance among multiple groups was determined using one-way ANOVA followed by Tukey's post hoc test. \*\*\*P<0.001; ns, not significant. **(d)** Relative fluorescence decay rate (RFDR) of the RNA aptamers under light exposure for 5 hours. Error bars indicate mean±SD (n=5). **(e)** Thermal stability of the RNA aptamers in vitro. $T_m$ represents the temperature at which fluorescence intensity decreases to 50% of the initial value. Error bars represent mean±SD (n=5). Statistical significance among multiple groups was determined using one-way ANOVA followed by Tukey's post hoc test. \*\*P<0.01; ns, not significant. **(f-g)** Fluorescence intensity of the RNA aptamers under different concentrations of KCl and $MgCl_2$. Error bars represent mean±SD (n=3). **(h)** Enzymatic resistance of the RNA aptamers. Error bars represent mean±SD (n=3).

Cas9 system (RAA/Cas9) where the engineered RNA aptamers functions as a transcriptional reporter, an alternative of traditional fluorescent protein, when appended to 3'UTR of *Cas9* (Fig 2a). So that, the post-transcriptional Cas9 mRNA could visualized, thereby bypassing the effect of fluorescent proteins on Cas9 activity (Fig 2a).

To validated this concept, we introduced these RAA/Cas9 constructs into *N. benthamiana* epidermal cells via Agrobacterium-mediated transformation. Fluorescence imaging of epidermal cells revealed that GFP and three RNA aptamers could fluorescently report *Cas9* expression (Fig 2b).

Next, we systematically assessed effects of RNA aptamers and *GFP* on *Cas9* expression at both transcriptional (RT-qPCR) and translational (immunoblotting) levels. Quantitative reverse transcription PCR (RT-qPCR) demonstrated equivalent Cas9 mRNA levels for four different constructs (Fig 2c). In contrast, Western blot analysis revealed that free Cas9 protein levels were lower in the P2A-GFP constructs compared with the RNA aptamer constructs (Fig 2d). The 3WJ-4×Bro construct exhibited Cas9 protein levels comparable to those of untagged Cas9, indicating that this configuration did not substantially alter protein accumulation.

## 3WJ-4×Bro/Cas9 system allows fast fluorescence-based selection of positive $T_1$ transformants

To evaluate the performance of the RAA/Cas9 system for rapid selection of positive transformants, we selected *AtTT4* as the target gene and constructed both RAA/Cas9 and GFP/Cas9 vectors, which were subsequently introduced into *Arabidopsis thaliana* via the floral dip method. Transgenic $T_1$ seedlings were screened on hygromycin-containing MS agar medium, followed by fluorescence detection and genomic PCR confirmation (Fig 3a). For the GFP/Cas9 construct, 32 out of 60 hygromycin-resistant seedlings exhibited fluorescence and were confirmed as positive transgenic plants by genomic PCR. However, 24 out of 28 non-fluorescent seedlings were also positive, indicating a 40% omission rate in identifying positive transformants using GFP fluorescence alone. For the 3WJ-4×Bro/Cas9 construct, 42 of 64 hygromycin-resistant seedlings exhibited fluorescence and were confirmed as positive, while 12 of 22 non-fluorescent seedlings were also positive, resulting in an omission rate of 18.75%. Similarly, the 3WJ-8×Bro/Cas9 and 3WJ-12×Bro/Cas9 constructs showed omission rates of 26.67% and 46.66%, respectively (Fig 3b and 3c). Additionally, the 3WJ-4×Bro/Cas9 construct exhibited the highest transformation efficiency (6.3%), followed by GFP/Cas9 (5%), 3WJ-12×Bro/Cas9 (4.6%), and 3WJ-8×Bro/Cas9 (4.2%) (Fig 3d). These results indicate that 3WJ-4×Bro is a promising alternative to GFP as a reporter in the Cas9 system.

Next, we investigated whether *Cas9* expression levels in transgenic plants could be assessed using fluorescence intensity. Fluorescence imaging of transgenic leaves showed that 3WJ-4×Bro exhibited the highest fluorescence intensity, followed by 3WJ-8×Bro, 3WJ-12×Bro, and GFP (Fig 3e and 3f), well in line with RT-qPCR results (Fig 3g), indicating that fluorescence intensity of RNA aptamers roughly reflects expression levels. This fluorescence-based visualization offered a convenient means to screen for transgenic lines with high *Cas9* expression, which is highly related to the editing efficiency. In addition, we examined whether RNA aptamers had adverse effects on transgenic plant growth. Phenotypic analysis at the seedling stage showed that neither RNA aptamers/Cas9 nor GFP/Cas9 constructs significantly affected *Arabidopsis* growth (Figs 3h and S2).

## Efficient generation of gene-edited $T_1$ plants using RAA/Cas9

To compare the editing efficiencies of the RAA/Cas9 systems with the GFP/Cas9 system in $T_1$ transgenic *Arabidopsis*, we extracted genomic DNA from transgenic lines and amplified sequences flanking each sgRNA target site using high-fidelity PCR. The resulting amplicons were sequenced via Sanger sequencing to evaluate genome editing outcomes. Genotyping of the edited plants revealed a variety of mutation types, including single-nucleotide deletions, insertions, and substitutions (Fig 4a). For the GFP-P2A/Cas9 construct, only 21% (12 out of 56) of $T_1$ plants had detectable mutations, including 4 plants with nucleotide insertions, 3 with deletions, and 5 with substitutions. For the 3WJ-4×Bro/Cas9 construct, a significantly higher mutation rate was observed, with 37.5% (21 out of 56) of $T_1$ plants showing detectable mutations, including

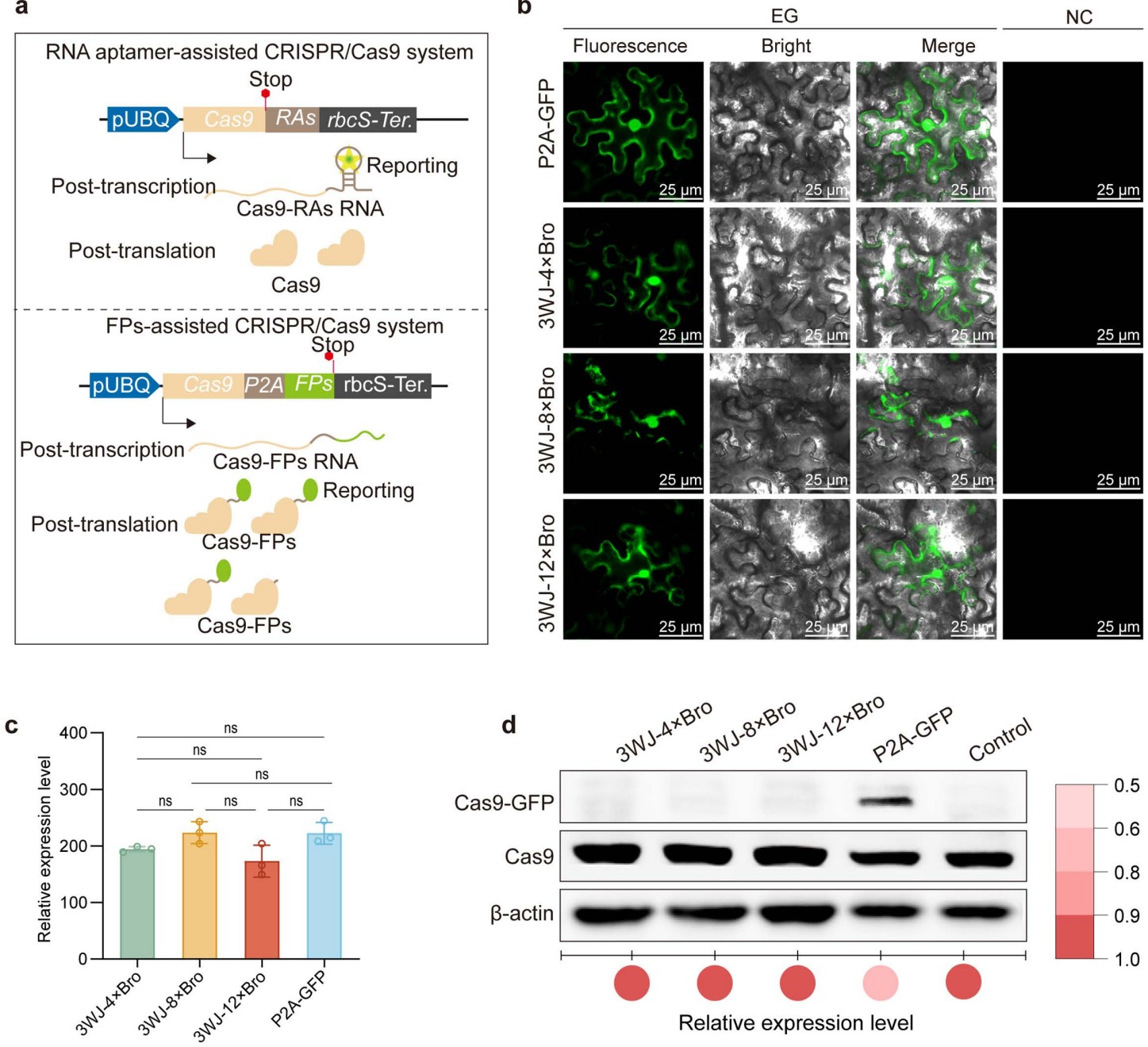

**Fig 2. The performance of RAA/Cas9 system. (a)** Schematic view of differences between RAs/Cas9 systems, in which RNA aptamers (3WJ-4×Bro, 3WJ-8×Bro, and 3WJ-12×Bro) are used as reporters, and the FPs/Cas9 system, in which GFP represents fluorescent proteins. **(b)** Representative image showing RAA and GFP to report Cas9 expression in tobacco leaf epidermal cells. Scale bar = 25 μm. **(c)** Effect of the RAA/Cas9 system on Cas9 gene expression at the transcriptional level. Relative expression levels of the Cas9 gene in tobacco leaf epidermal cells. Error bars indicate mean ± SD (n = 3). Statistical significance among multiple groups was determined using one-way ANOVA followed by Tukey's post hoc test. *$P < 0.05$, **$P < 0.01$; ns, not significant. **(d)** Western blot analysis of Cas9 protein expression levels, with β-actin used as a loading control. Relative expression level was calculated by the normalized grayscale values (compered to control) from three independent experiments. The color intensity of the bubbles indicates the relative expression level.

PLOS Genetics

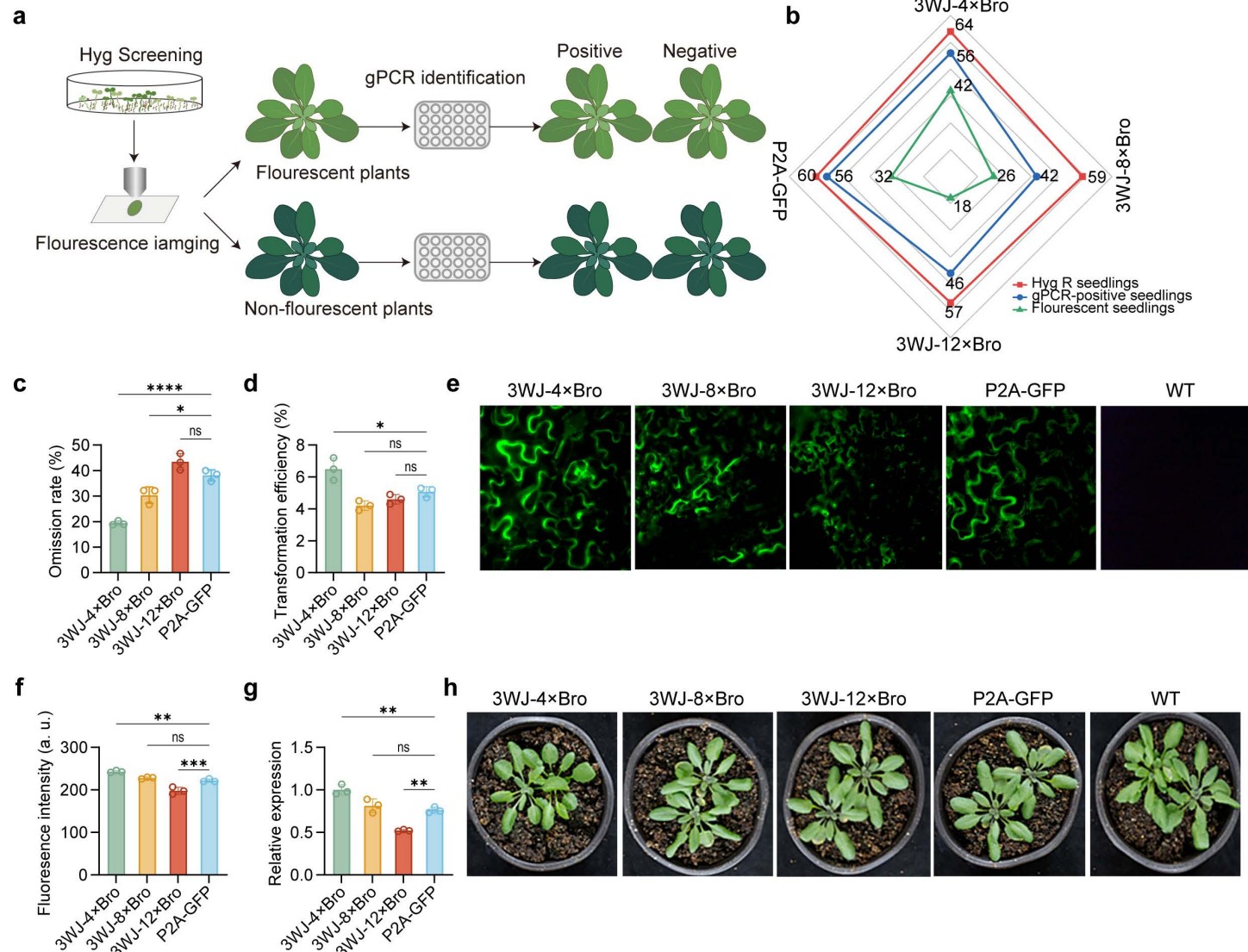

**Fig 3. The RAA/Cas9 system for positive transformants selection of T₁ *Arabidopsis* seedling. (a)** Schematic view of screening process of T₁ positive transgenic seedlings. **(b)** Number of T₁ transgenic *Arabidopsis* plants for four reporter constructs. Each axis represents a reporter construct, and colored lines indicate different detection methods (see legend). Data points represent the counted number of transgenic plants. **(c)** Omission rate of different reporters in positive transgenic plants selection. Omission rate was calculated by a ratio of the number of positive transformants identified in non-fluorescent plants to total T₁ plants number. **(d)** Transformation efficiency of gene-editing constructs with different reporters. **(e)** Fluorescently reporting expression level of *Cas9* by different reporter in T₁ transgenic *Arabidopsis* seedlings. **(f)** quantification of fluorescence in (e) by image J software. **(g)** The relative expression level of *Cas9* in T₁ transgenic plants quantified by RT-qPCR. Data are presented as mean ± SEM. Statistical significance among multiple groups was determined using one-way ANOVA followed by Tukey's post hoc test. **P < 0.01, ***P < 0.001; ns, not significant. **(h)** Growth performance of T₁ *Arabidopsis* seedlings.

7 with insertions, 9 with deletions, and 5 with substitutions, indicating greater editing efficiency than the GFP/Cas9 construct. The 3WJ-8×Bro/Cas9 and 3WJ-12×Bro/Cas9 constructs exhibited lower mutation rates than the 3WJ-4×Bro/Cas9 construct, at 28.57% and 19.57%, respectively (Fig 4b). Notably, only one T₁ plant from the 3WJ-4×Bro/Cas9 construct exhibited a homozygous mutation at the target site (denoted 4b#1), while no homozygous mutants were

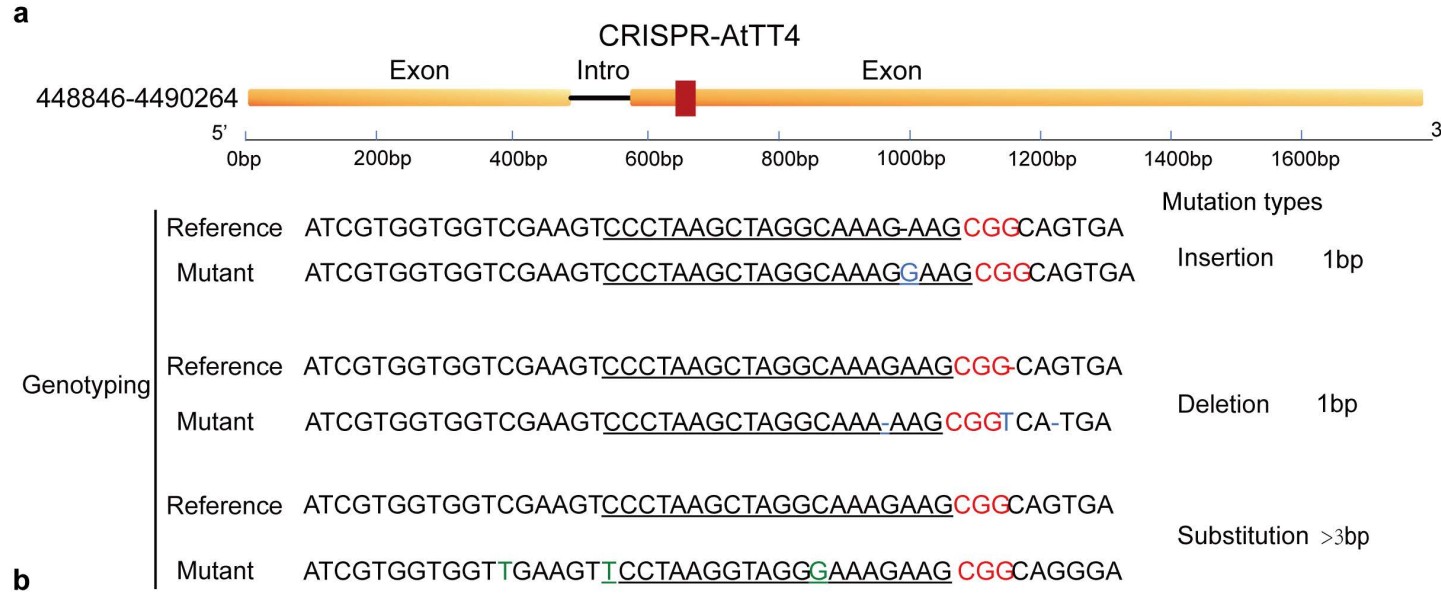

**a**

CRISPR-AtTT4

448846-4490264    Exon | Intro | Exon

5'    0bp  200bp  400bp  600bp  800bp  1000bp  1200bp  1400bp  1600bp    3'

Genotyping

| | Reference | ATCGTGGTGGTCGAAGTCCCTAAGCTAGGCAAAG-AAGCGGCAGTGA | Mutation types |
| | Mutant | ATCGTGGTGGTCGAAGTCCCTAAGCTAGGCAAAGGAAGCGGCAGTGA | Insertion  1bp |
| | Reference | ATCGTGGTGGTCGAAGTCCCTAAGCTAGGCAAAGAAGCGG-CAGTGA | |
| | Mutant | ATCGTGGTGGTCGAAGTCCCTAAGCTAGGCAAA-AAGCGGTCA-TGA | Deletion  1bp |
| | Reference | ATCGTGGTGGTCGAAGTCCCTAAGCTAGGCAAAGAAGCGGCAGTGA | |
| | Mutant | ATCGTGGTGGTTGAAGTTCCTAAGGTAGGGAAAGAAG CGGCAGGGA | Substitution >3bp |

**b**

| Constructs | T1 plants | Mutation types | | | Mutation Rate | Homozygous mutation | Homozygous mutation rate |
|---|---|---|---|---|---|---|---|
| | | Insertion | Deletion | Substitution | | | |
| GFP-P2A/Cas9 | 56 | 4 | 3 | 5 | 21.43% | 0 | 0 |
| 3WJ-4×Bro/Cas9 | 56 | 7 | 9 | 5 | 37.5% | 1 | 1.78% |
| 3WJ-8×Bro/Cas9 | 42 | 5 | 4 | 3 | 28.57% | 0 | 0 |
| 3WJ-12×Bro/Cas9 | 46 | 2 | 5 | 2 | 19.57% | 0 | 0 |

**c**

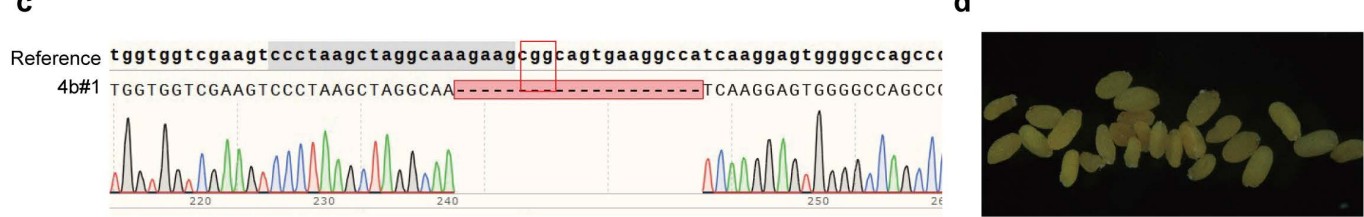

**d**

**Fig 4. Analysis of gene-editing efficiency of the RAA/Cas9 system in T₁ transgenic plants. (a)** Genotyping of T₁ edited plants. The PAM sequences are highlighted in red, and the sgRNA sequences are underlined. **(b)** statistics of mutation in T₁ generation. **(c)** Sanger sequencing chromatograms of homozygous edited lines 4b#1. The red box indicates the PAM sequence, the gray region marks the sgRNA sequence, red dashed lines represent deletions **(d)** Seed coat phenotypes of homozygous edited lines 4b#1.

detected in T₁ plants from other constructs (Fig 4c). In addition, all seeds from 4b#1 exhibited a white seed coat caused by the null mutant *tt4*, with no trait segregation, further confirming the presence of *tt4* alleles in 4b#1 (Fig 4d). In summary, the 3WJ-4×Bro/Cas9 construct demonstrated high efficiency in generating gene-edited plants in the T₁ generation, not only shortening the experimental timeline but also reducing costs, thereby offering significant advantages for practical CRISPR-based genome editing applications.

## Efficient isolation of Cas9-free mutants in the $T_2$ generation using the 3WJ-4×Bro/Cas9 system

Next, we performed a fluorescence-based visual screen on 100 $T_2$ *Arabidopsis* plants derived from individual $T_1$ gene-edited lines to identify Cas9-free mutants. From the GFP/Cas9 system, we obtained 23 Cas9-free $T_2$ plants (non-fluorescent). Among them, 13.04% (3 of 23) carried homozygous mutations at the *tt4* target site, 30.43% (7 of 23) carried heterozygous mutations, and 56.3% were wild-type. In the 3WJ-4×Bro/Cas9 system, 16% (4 of 25) of Cas9-free $T_2$ plants carried homozygous mutations at the *tt4* target site, 40% (10 of 25) carried heterozygous mutations, and 44% were wild-type. For the 3WJ-8×Bro/Cas9 and 3WJ-12×Bro/Cas9 systems, we identified 13.63% and 10% Cas9-free $T_2$ plants with homozygous mutations, and 25% and 30.4% with heterozygous mutations, respectively. The sorting efficiency of the 3WJ-4×Bro/Cas9 system for Cas9-free mutants in the $T_2$ generation reached 0.56%, representing a 30.2% increase compared to the GFP/Cas9 system (Table 1). These results indicate that the 3WJ-4×Bro/Cas9 system enables efficient selection of Cas9-free mutants in the $T_2$ generation.

To evaluate the performance of 3WJ-4×Bro/Cas9 in dual-target gene editing, we selected *AtTTG1* as the target gene (Fig 5a). Mutation of *AtTTG1* leads to trichome defects in leaf epidermal cells. Therefore, homozygous mutant lines in the $T_1$ generation can be rapidly identified based on leaf phenotypes. Both GFP/Cas9 and 3WJ-4×Bro/Cas9 generated homozygous mutant lines (Fig 5b). Further Sanger sequencing revealed that 3WJ-4×Bro/Cas9 induced homozygous mutations at both target sites, whereas the GFP/Cas9 system produced a homozygous mutation only at target site 1 (Fig 5c). Genotyping of $T_1$ transgenic plants showed that the 3WJ-4×Bro/Cas9 system exhibited mutation rates of 80% and 71% and homozygous mutation rates of 7% and 4% at the two target sites, all higher than the corresponding mutation (64% and 41%) and homozygous mutation rates (2% and 0%) of the GFP/Cas9 system (Fig 5d). In addition, fluorescence-based screening of 100 $T_2$ plants revealed that 40% of nonfluorescent $T_2$ 3WJ-4×Bro/Cas9 plants were homozygous Cas9-free mutants at target site 1 and 12% at target site 2, whereas the corresponding frequencies in GFP/Cas9 plants were only 26% and 6%, respectively (Fig 5e and Table 2). Moreover, only one $T_2$ 3WJ-4×Bro/Cas9 plant with homozygous Cas9-free mutations at both target sites 1 and 2 was identified (Table 2). Collectively, these results demonstrate that the 3WJ-4×Bro/Cas9 system has significantly greater potential for gene-editing applications than the traditional GFP/Cas9 system.

## Discussion

CRISPR/Cas genome editing technology has become an essential tool for both basic research and crop genetic improvement, with wide applications in enhancing yield [33,34], improving quality [35,36], and more. Among these systems, CRISPR/Cas9 relies on the Cas9 protein to induce targeted DNA cleavage. However, sustained expression of Cas9 can lead to off-target effects or cytotoxicity [37]. In plant genome editing, stable transformation is often employed to integrate the editing cassette into the host genome, followed by selection of the desired mutant lines and subsequent segregation to remove the Cas9 cassette, ultimately yielding non-transgenic mutant plants [22]. This process, however, requires repeated screening for transgene-positive and -negative individuals, which is labor-intensive and time-consuming.

**Table 1. Genotyping of Cas9-free $T_2$ transgenic plants.**

| Constructs | Non-fluorescent plants | Cas9-free mutations | | Sorting efficiency |
|---|---|---|---|---|
| | | Homozygous | Heterozygous | |
| 3WJ-4×Bro/Cas9 | 25 | 4 | 10 | 0.56% |
| 3WJ-8×Bro/Cas9 | 22 | 3 | 8 | 0.50% |
| 3WJ-12×Bro/Cas9 | 20 | 2 | 5 | 0.35% |
| GFP/Cas9 | 23 | 3 | 7 | 0.43% |

The sorting efficiency is calculated by dividing the proportion of mutant plants among non-fluorescent plants by the total 100 $T_2$ plants.

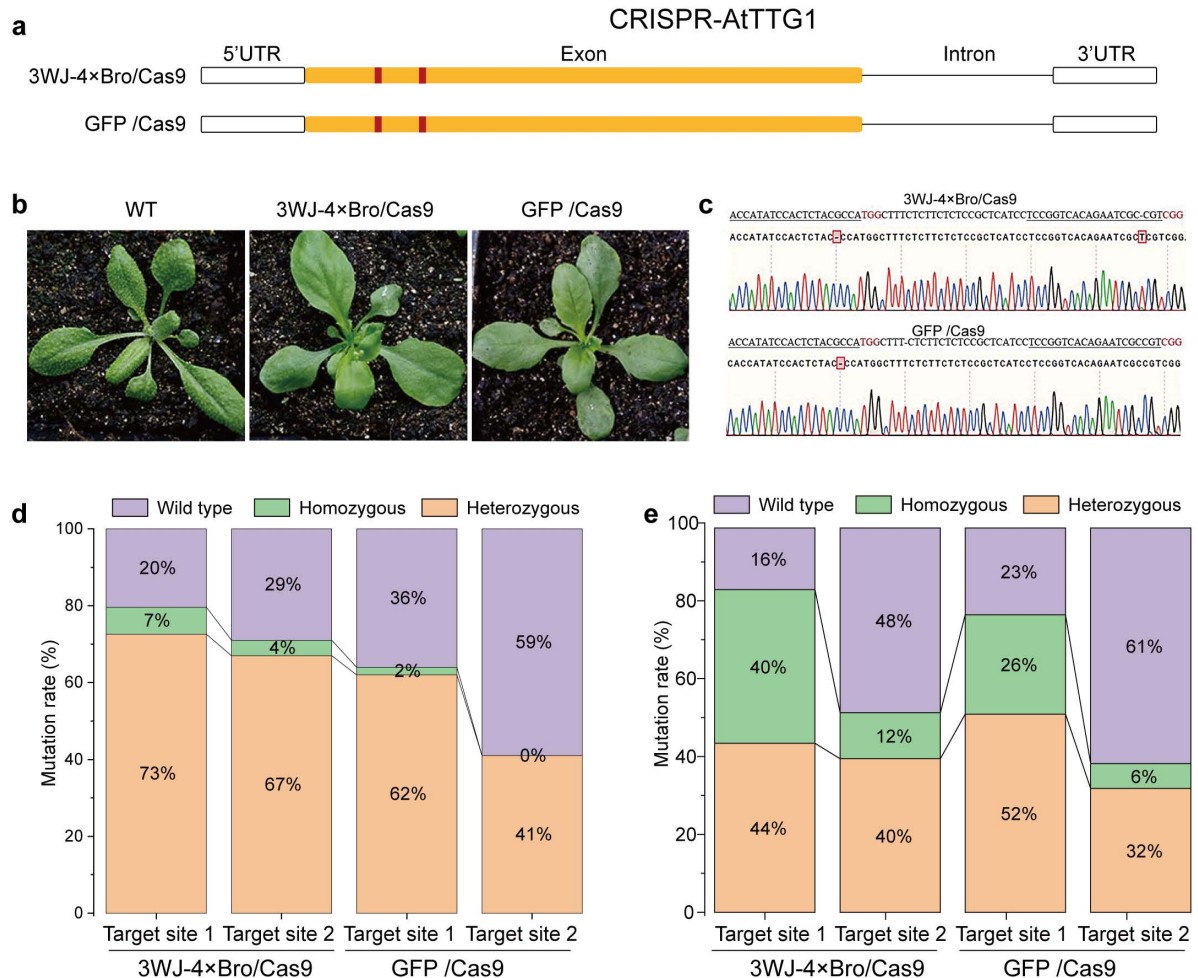

**Fig 5. Performance of 3WJ-4×Bro/Cas9 in double-target gene-editing.** (a) Schematic representation of target sites in *AtTTG1*. Target sites are indicated in red. Black boxes indicate exons, horizontal lines indicate introns, and open boxes indicate UTRs. (b) Edited phenotypes of T₁ transgenic plants. (c) DNA sequences of T₁ transgenic plants at target sites. (d) Stacked bar chart showing the proportions of different mutation types in T₁ transgenic plants. (e) Stacked bar chart showing the proportions of different mutation types in T₂ Cas9-free plants.

**Table 2. Genotyping of Cas9-free T₂ transgenic plants.**

| Constructs | T₂ non-fluorescent plants | Mutations T₂ plants without fluorescence | | | | Site 1and 2 |
| | | Target site 1 | | Target site 2 | | |
| | | Homozygous | Heterozygous | Homozygous | Heterozygous | Homozygous |
| --- | --- | --- | --- | --- | --- | --- |
| 3WJ-4 × Bro/Cas9 | 25 | 10 | 11 | 3 | 10 | 1 |
| GFP/Cas9 | 31 | 8 | 16 | 2 | 10 | 0 |

Therefore, developing a rapid and efficient marker system for identifying Cas9-expressing plants is of great significance for improving the practicability and applicability of plant genome editing.

RNA fluorescent aptamers offer a protein-independent labeling system that does not introduce exogenous proteins and has minimal interference with target gene expression. While RNA aptamers have been widely used for live-cell imaging in prokaryotic and mammalian systems, their application in plants is still in its early stages. Recent studies have

demonstrated that the 3WJ-4×Bro aptamer can be used as a fluorescent reporter for short transcripts in plants [32]. Based on the 3WJ-4×Bro structure, we constructed 3WJ-8×Bro and 3WJ-12×Bro by tandemly linking multiple Bro units, and systematically evaluated their fluorescence intensity, stability, and nuclease resistance. In vitro imaging showed a nonlinear enhancement of fluorescence with increasing Bro units, with signal saturation and a slight fluorescence inhibition observed at 12 units, suggesting that the RNA scaffold has a limited capacity for Bro aggregation. Excessive Bro units may impose spatial constraints that reduce overall fluorescence intensity. Further analyses revealed that these aptamers maintain fluorescence under low-salt conditions and resist nuclease degradation, indicating that their scaffolds achieve a favorable balance between structural rigidity and functional accessibility of Bro units, which is critical for consistent performance in cellular environments. Variations in fluorescence decay rates among the constructs indicate that photostability is influenced not only by scaffold length but also by folding conformation and local interactions between Bro units. Notably, 3WJ-4×Bro, despite having the shortest scaffold, exhibited an intermediate decay rate, highlighting the importance of structural packing. These findings are consistent with previous studies showing that RNA scaffold architecture and multimeric aptamer assemblies modulate fluorescence properties [32,38–42]. Transient expression assays in plants confirmed that the RNA aptamers can serve as effective fluorescent reporters in plant cells, and that the DFHBI-1T dye permeates well with minimal background fluorescence. Their robust stability and brightness make them promising tools for long-term imaging, live-cell studies, and investigation of RNA–protein interactions, as well as potential use in fluorescent sensors and in vitro assays.

In the context of CRISPR-Cas9 integration, the size and structure of the fluorescent marker system may affect Cas9 expression and cleavage efficiency. In this study, we integrated the RNA aptamer system (RNA aptamers) with CRISPR-Cas9 and compared its performance with a P2A-GFP-based system in *Arabidopsis*, evaluating *Cas9* expression reporting, editing efficiency, and the screening of Cas9-free edited lines. The results showed that RAA/Cas9 can effectively report *Cas9* expression status. However, in some T$_1$ transgenic lines, no fluorescence was detected, likely due to positional effects of random T-DNA insertion affecting transgene expression. High-expressing lines exhibited strong fluorescence, while those with low expression showed weak or undetectable signals. Fluorescence detection efficiency was significantly higher in lines expressing 3WJ-4×Bro and 3WJ-8×Bro than in those labeled with P2A-GFP, while the 3WJ-12×Bro system had the lowest detection rate. Editing efficiency was highest in the 3WJ-4×Bro/Cas9 system, followed by 3WJ-8×Bro/Cas9 and GFP/Cas9, with 3WJ-12×Bro/Cas9 showing the lowest editing efficiency. These differences may be attributed to several factors: (1) post-transcriptional factors, such as protein stability or folding efficiency, may contribute to the reduced accumulation of Cas9 in P2A-GFP constructs. GFP fluorescence depends on post-translational maturation, and the large size of Cas9 may hinder proper GFP folding, further reducing signal. Although the P2A peptide mediates co-expression and separation, cleavage is not always complete, and subtle effects on protein levels cannot be excluded, even though no unprocessed fusion protein was detected. Further studies, such as protein half-life or folding assays, are needed to clarify the underlying mechanisms; (2) the longer length and complex secondary structures of 3WJ-8×Bro and especially 3WJ-12×Bro (up to 883 bp) may hinder exposure of fluorophore-binding sites, weakening signal strength; (3) long exogenous RNAs in plant cells may trigger RNA silencing pathways, leading to aptamer degradation or unintended gene silencing, potentially affecting plant physiology. Thus, larger aptamers may be less suitable for use in plant CRISPR-Cas9 reporter systems.

In the screening of Cas9-free edited lines, both the RAA/Cas9 and GFP/Cas9 systems successfully identified Cas9-free edited plants, confirming that RNA aptamers are reliable markers for monitoring *Cas9* expression and guiding precise selection. Notably, in both single- and dual-target editing contexts, the Cas9–3WJ-4×Bro system exhibited higher sorting efficiency than GFP/Cas9 and was capable of generating homozygous Cas9-free mutations at both target sites. These results suggest that the 3WJ-4×Bro system holds potential for applications in Cas9-free selection protocols.

Beyond these experimental results, the superior performance of 3WJ-4×Bro can be attributed to its biochemical features, including strong thermal stability, nuclease resistance, and efficient folding that ensures dye accessibility in vivo.

PLOS Genetics

Its compact scaffold provides a favorable balance between fluorescence brightness and editing efficiency, whereas larger constructs such as 3WJ-12×Bro suffer from steric hindrance and reduced functional output. Functioning at the RNA level without requiring translation, aptamers impose minimal metabolic burden and enable rapid, real-time readouts of transcriptional activity. This makes them especially suitable for high-throughput identification of Cas9-free plants and highlights their broader potential in multiplex genome editing, live-cell RNA imaging, synthetic biology, and biosensing in plants.

In summary, RNA aptamers exhibit strong thermal stability, nuclease resistance, and low salt dependence in vitro. When combined with CRISPR-Cas9 in plants, the RAA/Cas9 system effectively reports *Cas9* expression. Compared to conventional protein-based reporters such as GFP, the CRISPR-Cas9 system assisted by 3WJ-4×Bro offers significant advantages in enhancing editing efficiency, optimizing transgenic line selection, and facilitating visual identification of Cas9-free edited plants. Therefore, the 3WJ-4×Bro RNA aptamer-assisted CRISPR/Cas9 system holds great potential for plant genome editing and may drive further applications in agricultural research and crop breeding.

## Materials and methods

### Plant material and growth conditions

In this study, wild-type *Arabidopsis* Columbia-0 was used. Seeds were sterilized with 5% NaClO for 20 minutes and washed 5 times with ddH$_2$O to remove residual disinfectant. The sterilized seeds were sown on 1/2 MS solid medium (2.2 g·L$^{-1}$Murashige Skoog powder, 1.5% sucrose, 10 g·L$^{-1}$agar, pH 5.8). After stratification for 2–3 days at 4°C, the seeds were transferred to a growth chamber. 7-day-old seedlings were transplanted into pots filled with a mixture of nutrient soil and vermiculite (3:1, v/v). The culture room conditions were maintained at 23°C under long-day photoperiods (16 h light/8 h dark) with a light intensity of 100–150 μmol·m$^{-2}$·s$^{-1}$.

### *In vitro* transcription and physicochemical assay of aptamers

Based on the previously constructed 3WJ-4×Bro module in our laboratory, we designed tandemly repeated constructs by inserting multiple Bro units into the 3WJ scaffold, resulting in 3WJ-8×Bro and 3WJ-12×Bro. The secondary structures of these RNA modules were predicted using minimum free energy (MFE) calculations via the RNAfold web server (http://rna.tbi.univie.ac.at/cgi-bin/RNAWebSuite/RNAfold.cgi).The synthesized sequences of 3WJ-8×Bro and 3WJ-12×Bro were cloned downstream of the T7 promoter in the pBluescript II SK(+) plasmid using *Kpn*I and *Sca*I double digestion, generating the recombinant vectors pBluescript II SK(+)-3WJ-8×Bro and pBluescript II SK(+)-3WJ-12×Bro. The recombinant plasmids were then extracted and purified. Using the laboratory-preserved plasmid PGEM-T-easy-3WJ-4×Bro and the newly constructed pBluescript II SK (+)-3WJ-8×Bro and pBluescript II SK (+)-3WJ-12×Bro as templates, PCR amplification was performed with M13F and M13R primers. Target fragments were recovered and subsequently used as DNA templates for in vitro transcription.

The *in vitro* transcription reaction was performed using the In vitro Transcription T7 Kit according to the manufacturer's protocol. Briefly, a 20 μL reaction mixture contained 1 μL DNA template, 2 μL 10×reaction buffer, 2 μL ATP/CTP/GTP/UTP mix, 2 μL T7 RNA polymerase mix, 0.5 μL RNase inhibitor and nuclease-free water. The reaction was incubated at 37°C for 2 hours. Following transcription, the reaction was treated with 15U RNase free DNase at 42°C for 30 min to remove the DNA template. The quality of transcribed RNA was analyzed by agarose gel electrophoresis, and the concentration was measured using a NanoDrop spectrophotometer at A260/A280. Only RNA with an A260/A280 ratio between 1.8 and 2.1 was used for downstream applications. A 50 μL reaction mixture was used for fluorescence imaging and intensity measurement, containing 50 μM aptamer RNA, 50 mM KCl, 5 mM MgCl$_2$, and 60 μM DFHBI-1T ((Z)-4-(3,5-difluoro-4-hydroxybenzylidene)-2-methyl-1-(2,2,2-trifluoroethyl)-1H-imidazol-5(4H)-one). The mixture was incubated at room temperature for 10 min. For fluorescence imaging, each aptamer-DFHBI-1T complex was visualized using a Blue Light Gel Imager [25]. The fluorescence intensity was measured at an excitation wavelength of 488 nm and an emission wavelength of 527 nm.

To prepare the aptamer RNA-DFHBI-1T complex, solutions were mixed to a final concentration of 4 µM RNA, 50 mM KCl, 5 mM MgCl$_2$, and 20 µM DFHBI-1T. Melting curve analysis was conducted using a real-time fluorescence PCR instrument. The temperature program was set from 25°C to 95°C, increasing by 0.5°C every 10 seconds, with a 30-second hold at each step before fluorescence signal acquisition. The melting curves and corresponding T$_m$ values were automatically calculated using the Bio-Rad software on the CFX96 system. To evaluate the fluorescence stability of the RNA-DFHBI-1T complex, the initial fluorescence intensity (Fi) was recorded using a fluorescence spectrophotometer. The samples were then exposed to fluorescent lighting (1500 lx) at room temperature for 5 hours. Residual fluorescence intensity (Fn) was measured once every hour. The relative fluorescence decay rate (RFDR) was calculated using the formula:

$$RFDR = \frac{Fi - Fn}{Fi} \times 100\% (n = 1 \sim 5)$$

For the RNase resistance assay, the aptamer RNA-DFHBI-1T complex was incubated at room temperature, followed by the immediate addition of 0.001 units of RNase A. Fluorescence intensity was continuously monitored using a multi-mode plate reader (Varioskan Flash) at 25°C, with an excitation wavelength of 488 ± 20 nm and an emission wavelength of 527 ± 20 nm. The fluorescence signal of 3WJ-n × Bro constructs was recorded every minute for 15 minutes. To assess the ion concentration dependence of the RNA-DFHBI-1T complex, 4 µM aptamer RNA was mixed with varying concentrations of MgCl$_2$ (0, 1, 2, 4, 6, 8, 10 mM) or KCl (0, 10, 20, 40, 60, 80, 100 mM), and 20 µM DFHBI-1T in a final volume of 1 mL. Fluorescence intensity was then measured using the Varioskan Flash plate reader.

## Plasmid construction and transformation

To evaluate the reporting efficiency of different tags, the pUBQ10:Cas9-P2A-GFP plasmid was linearized by double digestion with *HindIII* and *BsrGI*. TGA-RNA aptamer fragments (TGA-RAs) were amplified using primer pairs 4B-F/R, 8-F/R, and 12-F/R, and cloned into the linearized plasmid via homologous recombination to generate the pCas9-RAs constructs. The *AtTT4* gene, which encodes chalcone synthase (CHS), a key enzyme in the flavonoid biosynthesis pathway, was selected as the target gene. Mutations in *AtTT4* lead to a transparent seed coat and pale yellow phenotype [43].Functional disruption of *AtTTG1* leads to the absence of trichomes on the leaf epidermis of *Arabidopsis* [44]. CRISPR/Cas9 target sites were designed using the CRISPR-P 2.0 online tool (http://crispr.hzau.edu.cn/CRISPR2/) to ensure high specificity and minimal off-target effects. The sgRNA sequences were selected based on the presence of NGG PAM motifs recognized by SpCas9. The sgRNA expression cassette was constructed by inserting the CRISPR target sequences downstream of the *AtU6-26* promoter using pAtU6-26 as the template, generating the intermediate plasmid pAtU6-26-*TT4*. Subsequently, pAtU6-26-*TT4* and the pUBQ10:Cas9-P2A-GFP or pCas9-RAs vectors were digested with *KpnI* and *SalI*. The sgRNA-containing fragment from pAtU6-26-*TT4* was ligated into the linearized *Cas9* expression vectors, resulting in the final constructs *pTT4*-Cas9-P2A-GFP and *pTT4*-Cas9-RAs. Plasmids were extracted using a Plasmid Mini Kit and introduced into *Agrobacterium tumefaciens* strain GV3101 by the freeze-thaw method. Transformed strains were selected on LB medium supplemented with 50 µg/mL kanamycin and 25 µg/mL rifampicin, and confirmed by PCR.

For transient expression, *Agrobacterium* strains carrying the CRISPR/Cas9-reporter constructs were cultured in LB medium with antibiotics at 28°C, 220 rpm for 16–18 h. The cultures were centrifuged at 4,000 × g for 10 min at 4°C, and the pellet resuspended in infiltration buffer (10 mM MgCl$_2$, 10 mM MES, 200 µM acetosyringone, pH 5.6) to an OD$_{600}$ of 0.8-1.0. After incubation at room temperature for 30 min, the suspension was infiltrated into the abaxial side of *N. benthamiana* leaves using a 1 mL needleless syringe. Plants were kept in the dark for 16 h and then grown under normal light conditions for 48 h. For RNA aptamer-based reporters, 60 µM DFHBI-1T was injected into the same infiltrated region, followed by incubation in the dark for 30 min before fluorescence microscopy. The *p*UBQ10:Cas9-P2A-GFP construct required no dye injection and was directly visualized.

For stable transformation, *Arabidopsis* plants at the flowering stage with multiple open flowers on the main inflorescence were used. *Agrobacterium* cultures were prepared as described, and the $OD_{600}$ was adjusted to 0.8-1.0 in floral dip buffer (MS liquid medium, 5% sucrose, 100 µM acetosyringone, 0.05% Silwet L-77). The inflorescences were dipped into the bacterial suspension for 30–60 seconds to ensure thorough coverage. After dipping, plants were covered with plastic wrap for 16–24 hours to maintain humidity, then grown under normal conditions until seed set and maturation.

## RT-qPCR

Total RNA was extracted from *N.benthamiana* leaves 48 hours after agroinfiltration using the Plant RNA Purification Kit (TIANGEN, China) according to the manufacturer's instructions. To minimize the impact of mosaic expression, leaf discs were harvested exclusively from regions exhibiting strong GFP fluorescence, thereby enriching for successfully transformed cells. For each biological replicate, leaf discs from three independent plants were pooled to prepare total RNA and total protein extracts. At least three independent biological replicates were analyzed. The concentration and purity of RNA were assessed with a NanoDrop 2000 spectrophotometer (Thermo Fisher Scientific). For each sample, 1 µg of total RNA was reverse transcribed into cDNA using the PrimeScript RT reagent Kit with gDNA Eraser (Takara, Japan) to remove genomic DNA contamination. RT-qPCR was performed using TB Green Premix Ex Taq II (Takara). The reaction program was as follows: 95 °C for 30 s, followed by 40 cycles of 95 °C for 5 s and 60 °C for 30 s. Each reaction was conducted in technical triplicates, and melting curve analysis was performed to verify amplification specificity. The relative expression level of Cas9 was calculated using the $2^{-\Delta\Delta Ct}$ method, with *NbACTIN* as the internal reference gene.

## Protein extraction and western blotting

Total protein was extracted from *N.benthamiana* leaves 48 hours after agroinfiltration. The method for collecting leaf disc samples is identical to that for RT-qPCR. Leaf tissues were ground in liquid nitrogen and lysed in NP-40 lysis buffer on ice for 10 minutes. The lysates were centrifuged at 12,000 × g for 15 minutes at 4 °C, and the supernatants were collected. Protein concentration was measured using the BCA Protein Assay Kit (Beyotime, China). Equal amounts of protein (20 µg per sample) were separated by 12% SDS-PAGE and transferred onto PVDF membranes. The membranes were blocked in 5% non-fat milk prepared in TBST buffer (20 mM Tris-HCl, 150 mM NaCl, 0.1% Tween-20) for 1 hour at room temperature. Subsequently, the membranes were incubated overnight at 4 °C with a mouse anti-FLAG monoclonal antibody to detect FLAG-tagged Cas9 proteins. After washing with TBST, the membranes were incubated with HRP-conjugated secondary antibodies for 1 hour at room temperature. Protein signals were detected using an enhanced chemiluminescence (ECL) reagent and visualized with a ChemiDoc MP Imaging System (Bio-Rad).

## Selection and identification of transgenic and gene-edited *Arabidopsis*

$T_0$ transgenic Arabidopsis lines were selected on MS medium containing 20 mg·L$^{-1}$ hygromycin. Genomic DNA was extracted from seedlings at the 4–5 leaf stage for PCR verification of transgene integration, and fluorescence microscopy was used to assess reporter expression in leaves. To identify gene-edited plants, genomic DNA from $T_1$ lines was amplified using primers flanking the sgRNA target site, and PCR products were subjected to Sanger sequencing. For the selection of Cas9-free mutants in the $T_2$ generation, seedlings were screened on MS medium supplemented with 30 mg·L$^{-1}$ hygromycin. Non-fluorescent individuals were identified via fluorescence microscopy. Genomic DNA was then extracted from these candidates, and PCR was performed to confirm editing events at the target locus.

## Data statistics and analysis

The data Data were processed and analyzed using GraphPad Prism and SPSS (SPSS Inc., Chicago, IL, USA). Sequencing data were analyzed using SnapGene, and fluorescence images were processed with Image J.

## Supporting information

**S1 Fig.  Minimum free energy (MFE) secondary structure of 3WJ-n × Bro fluorescent aptamer.** (a) MFE secondary structure of 3WJ-4 × Bro. (b) MFE secondary structure of 3WJ-8 × Bro.(c) MFE secondary structure of 3WJ-12 × Bro.
(DOCX)

**S2 Fig.  Growth status of transgenic *Arabidopsis* seedlings.** (a) Rosette leaf area. (b) Number of rosette leaves. Error bars represent mean ± SD (n = 10). Statistical significance among multiple groups was determined using one-way ANOVA followed by Tukey's post hoc test. ns, not significant.
(DOCX)

**S1 Table.  Primer sequences used in this study.**
(DOCX)

**S2 Table.  Minimal data set.**
(XLSX)

## Acknowledgments

We sincerely thank Dr. Haodong Chen of Peking University for generously providing the plasmid vectors essential to our experiments. His support greatly contributed to the success of this study.

## Author contributions

**Conceptualization:** Jiuyuan Bai, Yun Zhao.

**Funding acquisition:** Jiuyuan Bai, Yun Zhao.

**Investigation:** Sha Liu, Jiuyuan Bai, Bo Zhan.

**Methodology:** Sha Liu, Bo Zhan, Mengyue Dong.

**Supervision:** Yun Zhao.

**Visualization:** Sha Liu, Junyu Yao, Jiayu Zhang, Jia Yi, Qicong Li, Yucheng Shen.

**Writing – original draft:** Sha Liu.

**Writing – review & editing:** Yazhou Chen, Yun Zhao.

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
