## [Decision Letter · Decision Letter 0]

22 Aug 2025

PGENETICS-D-25-00713

Development of an RNA aptamer-assisted CRISPR/Cas9 system for efficiently generating and isolating Cas9-free mutants in plant

PLOS Genetics

Dear Dr. Zhao,

Thank you for submitting your manuscript to PLOS Genetics. After careful consideration, we feel that it has merit but does not fully meet PLOS Genetics's publication criteria as it currently stands. Therefore, we invite you to submit a revised version of the manuscript that addresses the points raised during the review process.

Please submit your revised manuscript within 30 days before Nov. 20, 2025. If you will need more time than this to complete your revisions, please reply to this message or contact the journal office at plosgenetics@plos.org. Please include the following items when submitting your revised manuscript:

We look forward to receiving your revised manuscript.

Kind regards,

Bao-Cai Tan

Academic Editor

PLOS Genetics

Quanjiang Ji

Section Editor

PLOS Genetics

Aimée Dudley

Editor-in-Chief

PLOS Genetics

Anne Goriely

Editor-in-Chief

PLOS Genetics

**Journal Requirements:**

At this stage, the following Authors/Authors require contributions: Sha Liu, Jiuyuan Bai, Bo Zhan, Junyu Yao, Jiayu Zhang, Jia Yi, Mengyue Dong, Qicong Li, Yucheng Shen, Yazhou Chen, and Yun Zhao. Please ensure that the full contributions of each author are acknowledged in the "Add/Edit/Remove Authors" section of our submission form.

The list of CRediT author contributions may be found here: https://journals.plos.org/plosgenetics/s/authorship#loc-author-contributions

- ® on page: 22.

3) Some material included in your submission may be copyrighted. According to PLOSu2019s copyright policy, authors who use figures or other material (e.g., graphics, clipart, maps) from another author or copyright holder must demonstrate or obtain permission to publish this material under the Creative Commons Attribution 4.0 International (CC BY 4.0) License used by PLOS journals. Please closely review the details of PLOSu2019s copyright requirements here: PLOS Licenses and Copyright. If you need to request permissions from a copyright holder, you may use PLOS's Copyright Content Permission form.

Potential Copyright Issues:

- Please confirm (a) that you are the photographer of Figure 3h, 4d, and 5b, or (b) provide written permission from the photographer to publish the photo(s) under our CC BY 4.0 license.

- Figure 3a. Please confirm whether you drew the images / clip-art within the figure panels by hand. If you did not draw the images, please provide (a) a link to the source of the images or icons and their license / terms of use; or (b) written permission from the copyright holder to publish the images or icons under our CC BY 4.0 license. Alternatively, you may replace the images with open source alternatives. See these open source resources you may use to replace images / clip-art:

4) We note that your Data Availability Statement is currently as follows: "All data are available in the main text or the supplementary materials. All materials (seed stocks, plasmids) are available upon request.". Please confirm at this time whether or not your submission contains all raw data required to replicate the results of your study. Authors must share the “minimal data set” for their submission. PLOS defines the minimal data set to consist of the data required to replicate all study findings reported in the article, as well as related metadata and methods (https://journals.plos.org/plosone/s/data-availability#loc-minimal-data-set-definition).

- The points extracted from images for analysis..

5) Please ensure that the funders and grant numbers match between the Financial Disclosure field and the Funding Information tab in your submission form. Note that the funders must be provided in the same order in both places as well.

State what role the funders took in the study. If the funders had no role in your study, please state: "The funders had no role in study design, data collection and analysis, decision to publish, or preparation of the manuscript.".

**Reviewers' comments:**

Reviewer's Responses to Questions

**Comments to the Authors:**

Reviewer #1: Manuscript "Development of an RNA aptamer-assisted CRISPR/Cas9 system for efficiently generating and isolating Cas9-free mutants in plant" by Liu et al. developed an RNA aptamer (3WJ-4×Bro)-based CRISPR/Cas9 system (3WJ-4×Bro/Cas9) to address the complexity and inefficiency of screening Cas9-free edited plants in plant genome editing. The study characterized the physicochemical properties of RNA aptamers in vitro, constructed the system, and validated its transformation efficiency, editing efficiency, and screening performance for Cas9-free mutants in Arabidopsis thaliana. The results showed that this system outperforms the conventional GFP/Cas9 system. The innovative application of RNA aptamers as transcriptional reporters in plant genome editing provides a new strategy for efficiently generating transgene-free edited plants, with significant theoretical and practical value. However, some result analyses require refinement. The following are some comments for discussion and consideration.

Question 1: The authors attribute the fastest fluorescence decay of 3WJ-12×Bro and the slowest decay of 3WJ-8×Bro to 'scaffold length and structural density. Is this interpretation speculative? If scaffold length were the sole determinant, why does 3WJ-4×Bro (shortest scaffold) not exhibit the slowest decay rate as one might expect? This apparent contradiction requires clarification. Are there any published literature reports were used in discussion

Question 2: The minimum Tm value of the aptamer is as high as 58 degrees, which indicates that the aptamers theoretically have well stability in the physiological environment of 37 degrees. The conclusion that the aptamers 'theoretically have good stability' is insufficiently rigorous. The Tm value solely reflects the melting temperature and does not directly demonstrate long-term functional stability at 37°C. We recommend that excessive subjective interpretations be avoided in the Results section; such discussions may be more appropriately addressed in the Discussion."

Question 3: Was a positive control included in the RNase A resistance assay (Fig 1h)? Please explicitly state the degradation conditions (enzyme concentration/duration).

Question 4: Figure 2a: "RAs" may be "RAAs", "FPs" may be "GFP".

Question 5: Cas9 Expression Discrepancy (Fig 2c–d)�While RT-qPCR shows equivalent mRNA levels, Western blot indicates reduced Cas9 protein for GFP fusions. The explanation ("incomplete 2A cleavage") lacks direct evidence (e.g., unprocessed fusion protein not shown).

Question 6: The claim that “the 3WJ-4×Bro/Cas9 system resulted in a 25–73% increase in mutation rates at single target sites and at least a 2.5-fold increase in homozygous mutation rates in the T1 generation compared to GFP/Cas9 (Fig 5d) lacks specific numerical support. We recommend presenting comparative data directly rather than using fold-change.

Question 7: Figure 5e is inconsistent with the results presented in Table 2.

We recommend revising the wording to reduce speculative phrasing in the Results section. Additionally, certain conclusions contain instances of overinterpretation; descriptions should be modified to clearly delineate the scope of applicability for these claims.

Reviewer #2: This manuscript presents an RNA aptamer-based CRISPR/Cas9 platform called 3WJ-4×Bro/Cas9 to facilitate efficient selection of Cas9-free, genome-edited plants. The authors’ system utilizes an RNA aptamer fused to the Cas9 transcript to report gene expression via fluorescence, avoiding interference with protein translation. While the manuscript introduces a potentially valuable tool, it should address the following:

Line 138: “Strikingly, immunoblotting revealed GFP fusion caused substantial attenuation of free Cas9 accumulation (Fig 2d).”

The authors should tone down this statement, because there seems to be not so much difference in signal strength between the 3WJ-8xBro and the GFP control.

Figure 3c and 3d: Are there any statistical indicators over the bar graphs? Related to this—line 164-167: 3WJ-4×Bro/Cas9 construct showing 5.6% transformation efficiency and GFP/Cas9 showing 5% do not seem to me as statistically significant difference.

Figure S2 is missing(?)

Regarding the assessment of genome editing efficiency by Sanger sequencing presented in the manuscript: While Sanger sequencing can reveal useful information about genome editing efficiency, particularly in homozygous, mono- or bi-allelic mutants, it does not provide an in-depth quantification (especially in heterozygous lines). To this end, have the authors considered using NGS for analysis of genome editing efficiency? This would provide more quantifiable data.

Reviewer #3: uploaded as attachment

**Have all data underlying the figures and results presented in the manuscript been provided?**

Reviewer #1: Yes

Reviewer #2: Yes

Reviewer #3: Yes

PLOS authors have the option to publish the peer review history of their article (what does this mean? ). If published, this will include your full peer review and any attached files.

**Do you want your identity to be public for this peer review?** For information about this choice, including consent withdrawal, please see our Privacy Policy .

Reviewer #1: No

Reviewer #2: No

Reviewer #3: No

**Figure resubmission:**
---

## [Decision Letter · Decision Letter 1]

19 Oct 2025

Dear Dr Zhao,

We are pleased to inform you that your manuscript entitled "Development of an RNA aptamer-assisted CRISPR/Cas9 system for efficiently generating and isolating Cas9-free mutants in plant" has been editorially accepted for publication in PLOS Genetics. Congratulations!

Yours sincerely,

Bao-Cai Tan

Academic Editor

PLOS Genetics

Quanjiang Ji

Section Editor

PLOS Genetics

Aimée Dudley

Editor-in-Chief

PLOS Genetics

Anne Goriely

Editor-in-Chief

PLOS Genetics

BlueSky: @plos.bsky.social

Comments from the reviewers (if applicable):

Reviewer's Responses to Questions

**Comments to the Authors:**

Reviewer #1: I have reviewed the authors' responses and the revised manuscript. All my concerns have been adequately addressed through the additional data and textual revisions.

Reviewer #3: Thank you for addressing the comments and providing clarifications

**Have all data underlying the figures and results presented in the manuscript been provided?**

Reviewer #1: None

Reviewer #3: Yes

PLOS authors have the option to publish the peer review history of their article (what does this mean? ). If published, this will include your full peer review and any attached files.

**Do you want your identity to be public for this peer review?** For information about this choice, including consent withdrawal, please see our Privacy Policy .

Reviewer #1: No

Reviewer #3: No

**Data Deposition**

http://datadryad.org/submit?journalID=pgenetics&manu=PGENETICS-D-25-00713R1

**Press Queries**

---

## [Editor Report · Acceptance letter]

PGENETICS-D-25-00713R1

Development of an RNA aptamer-assisted CRISPR/Cas9 system for efficiently generating and isolating Cas9-free mutants in plant

Dear Dr Liu,

We are pleased to inform you that your manuscript entitled "Development of an RNA aptamer-assisted CRISPR/Cas9 system for efficiently generating and isolating Cas9-free mutants in plant" has been formally accepted for publication in PLOS Genetics! Your manuscript is now with our production department and you will be notified of the publication date in due course.

With kind regards,

Anita Estes

PLOS Genetics

On behalf of:
